# Polar Metabolites Profiling of Wheat Shoots (*Triticum aestivum* L.) under Repeated Short-Term Soil Drought and Rewatering

**DOI:** 10.3390/ijms24098429

**Published:** 2023-05-08

**Authors:** Joanna Szablińska-Piernik, Lesław Bernard Lahuta

**Affiliations:** Department of Plant Physiology, Genetics and Biotechnology, University of Warmia and Mazury in Olsztyn, Oczapowskiego Street 1A/103A, 10-719 Olsztyn, Poland

**Keywords:** wheat, shoot, drought, carbohydrates, proline, malate

## Abstract

The response of wheat (*Triticum aestivum* L.) plants to the soil drought at the metabolome level is still not fully explained. In addition, research focuses mainly on single periods of drought, and there is still a lack of data on the response of plants to short-term cyclical periods of drought. The key to this research was to find out whether wheat shoots are able to resume metabolism after the stress subsides and if the reaction to subsequent stress is the same. Gas chromatography coupled with mass spectrometry (GC-MS) is one of the most valuable and fast methods to discover changes in the primary metabolism of plants. The targeted GC-MS analyses of whole shoots of wheat plants exposed (at the juvenile stage of development) to short-term (five days) mild soil drought/rewatering cycles (until the start of shoot wilting) enabled us to identify 32 polar metabolites. The obtained results revealed an accumulation of sugars (sucrose, fructose, glucose, and 1-kestose), proline, and malic acid. During five days of recovery, shoots regained full turgor and continued to grow, and the levels of accumulated metabolites decreased. Similar changes in metabolic profiles were found during the second drought/rewatering cycle. However, the concentrations of glucose, proline, and malic acid were higher after the second drought than after the first one. Additionally, the concentration of total polar metabolites after each plant rewatering was elevated compared to control samples. Although our results confirm the participation of proline in wheat responses to drought, they also highlight the responsiveness of soluble carbohydrate metabolism to stress/recovery.

## 1. Introduction

Wheat (*Triticum aestivum* L.) is one of the major crops cultivated worldwide and an important source of energy for the human diet and animal feed [1,2]. The yield of wheat may be dramatically decreased by drastic environmental and climatic changes, leading to an increasing frequency of severe drought conditions [3,4]. It is predicted that up to 60% of the current wheat-growing area will face simultaneous severe water scarcity events by the end of the twenty-first century [5,6]. Moreover, abiotic stresses, including drought, might occur individually or sequentially. Drought adversely affects crop growth, development, and yield via physiological and biochemical disturbances in whole plants. However, the negative effects are related to the drought intensity (rate of decrease in the water availability and duration of the water deficit), as well as plant species and developmental stages [7,8].

Water stress leads to weakening and inhibition of plant growth, disturbances in photosynthesis and carbon and nitrogen metabolism [9], as well as overproduction of reactive oxygen species [10]. The limited and decreasing availability of water in the soil, leading to soil drought, indicates changes in the stomata aperture in leaves to limit transpiration and water loss from the cells [11]. Water retention in cells is achieved through the accumulation of various osmolytes in the cytoplasm and vacuole—sugars, amino acids, inorganic ions, and organic acids—lowering the water potential [12]. However, prolonged water deficits and minimized gas exchange lead to a decrease in CO_2_ concentration in the leaf [13,14]. The decreased ratio of CO_2_ to O_2_ in chloroplasts results in the intensification of the photorespiration process [15,16]. It reduces the efficiency of CO_2_ assimilation and causes losses of nitrogen and energy. However, it can also be a partial CO_2_ recirculation pathway [17,18]. Limited water availability and reduced CO_2_ assimilation (in the Calvin–Benson cycle) lead to disturbances in the processes of electron transport, energy production, and maintaining redox balance in chloroplasts. As a result, there is excessive generation of reactive oxygen species, and the photosynthetic apparatus is damaged [19]. This causes disturbances in the course of various metabolic pathways and the export of photoassimilates from the leaves (as sources) to the receiving sink tissues (roots, stem, and generative organs) [20,21].

Apart from osmolytes [22,23], cells can also synthesize osmoprotectants (trehalose, glycine betaine, and polyols), maintaining membrane integrity and protecting macromolecule structure [24]. Accumulation of osmolytes and osmoprotectants affects both primary and secondary metabolism, as revealed in metabolomic studies of model plants, e.g., *Arabidopsis thaliana* [25] and *Medicago truncatula* [26], as well as major crops, e.g., maize [27], wheat [28], rice [29], barley [30], soybean [31], and pea [32,33,34]. Moreover, some metabolites accumulated under water deficit could be candidates for metabolic markers of stress, e.g., proline, found in various plant species [35], sucrose, citric acid, and phosphate in *Medicago truncatula* [12], and glutamine/glutamate in rice [36].

So far, research on wheat’s response to drought at the metabolome level has shown that limited water availability is associated with changes in osmotic regulation, energy production, and oxidative damage [37,38]. It has been demonstrated that sugar and nitrogen metabolism are mostly affected by water deficiency in the leaves of wheat [28,37,38,39]. Moreover, the drought-tolerant capacity of wheat seedlings was related to the accumulation of amino acids, organic acids, alkaloids, and flavonoids [40]. The important role of some metabolites in response to drought has also been revealed by metabolomic analyses carried out on other plant species (i.e., coffee, ryegrass) subjected to cyclic drought/rehydration treatment [8,41,42]. In wheat, it has been shown that not only the water deficit itself but also its frequency affects plant growth and secondary metabolism in flag leaf [43], as well as that the greatest changes are related to the accumulation of amino acids [11] and soluble sugars in leaf tissues [44]. Moreover, the drought tolerance of wheat seems to be related to sucrose metabolism [45,46].

Among various methods used in metabolomic studies [30,31], the gas chromatography-mass spectrometry (GC-MS) method seems to be valuable in monitoring changes in metabolic profiles in early response to drought [11,27,33,34]. In the present study, the GC-MS approach enabled us to compare changes in the polar metabolite profiles of wheat shoots (*Triticum aestivum* L.) during the plant’s exposure to repeated short-term (mild) soil droughts and after recovery from water deficit stress.

## 2. Results and Discussion

### 2.1. The Effect of Repeated Short-Term Soil Drought/Rewatering Cycles on Plant Growth and Development

During 20 days of vegetation (between the 14th and 34th day after sowing, DAS), well-watered (control) plants reached a height of about 30–35 cm, and shoots started tillering. The water content (WC) was stable at 6.5–7 g H_2_O g^−1^ dry weight (DW), which corresponded with 86–89% of fresh weight (Figure 1).

The fresh weight (FW) of shoots increased more than fourfold (Figure 2), and the width of fully expanded leaves was ca. 5–7 mm at mid-length (Figure 1). 

The duration of the first and second droughts was relatively short (until visible loss of turgor and shoot wilting), enabling fast recovery of shoot growth after plants rewatering. After five days of drought, the WC decreased from 7.96 to 3.33 g H_2_O g^−1^ DW, whereas during rewatering it increased to 6.86 g H_2_O g^−1^ DW (as high as in the control). Both short-term soil droughts caused a strong decrease in shoot FW (twofold, compared with the control) and inhibited shoot growth (Figure 1). However, after recovery from the second stress, plants regained turgor, continued to grow, and produced new leaves (Figure 1). The FW of the shoot increased more than twofold, but to a level still lower than in the control plants (Figure 2). Moreover, the shoot tillering was restricted (Figure 1). The above disturbances in the growth of wheat plants are consistent with previous reports on the response of plants to drought stress [47,48,49]. Growth inhibition caused by the soil drought results from disturbances in photosynthesis, transpiration, photorespiration, transport of water and mineral/organic compounds, as well as photoassimilated distribution between source and sink tissues [50,51]. In our work, we focused not on the measurement of changes in physiological processes but only on rearrangements in the metabolic profiles of shoots in wheat in response to repeated soil drought/rewatering.

### 2.2. Changes in Polar Metabolite Profiles during Control Plant (Well-Watered) Vegetation

The targeted GC-MS-based approach, using original standards as well, enabled us to identify 32 polar metabolites, classified into soluble carbohydrates, proteinogenic and non-proteinogenic amino acids, organic acids, and remaining compounds. The same groups as well as individual polar metabolites have previously been found in wheat leaves, kernels, or phloem exudate [52,53,54] as well as in wheat sprouts [55,56] using mass spectrometry. Changes in the concentrations of polar metabolites during plant vegetation (14–34 DAS) were reflected by a shift in the distribution of samples, revealed by principal component analysis (PCA). There were clear differences between the samples collected at the 14th DAS and those collected later (Figure 3A). The distribution of samples depended mainly on phosphoric acid, glucose, fructose, and sucrose (Figure 3B). 

Soluble carbohydrates were the major group of polar metabolites in the control plants, constituting 36–82% of total identified polar metabolites (TIPMs), whereas total organic acids (TOAs) shared ca. 6–11% and amino acids (proteinogenic and non-proteinogenic) ca. 7–12% (Table 1). 

The trends of changes in the concentration of TIPMs among the vegetation in the wheat shoots of control plants were determined mainly by changes in soluble carbohydrates. During intensive shoot growth, between 19 and 34 DAS (Figure 1), the concentrations of fructose, glucose, and sucrose (the predominant sugars) increased gradually up to 2.45, 10.70, and 7.63 mg g^−1^ DW, respectively. Additionally, 1-kestose, found in traces at the 19th DAS, increased up to the 34th DAS, but to a much lower level (0.14 mg g^−1^ DW, Table 1). This composition of sugars seems to be typical of wheat leaf tissues [57]. Moreover, 1-kestose is one of the major fructans present in monocotyledonous plants in temperate climates, among them wheat and other cereals [58]. They occur mainly in grains, fruits, and vegetables and are used as storage carbohydrates for growing seedlings [59]. 

Changes in TIPMs in wheat were also influenced by changes in total organic acids (TOAs) and total proteinogenic amino acids (TPAAs), whose concentrations significantly decreased with the growth of shoots (Table 1). The dominant proteinogenic amino acids were asparagine, aspartic acid, and serine, whereas among organic acids, malate and citrate dominated. However, their content was relatively low (<2.5 mg g^−1^ DW, Table 1). Tissues also contained small amounts of hydroxyproline, γ-aminobutyric acid (GABA), and urea (Table 1). Additionally, tissues at the 14th DAS contained a considerable amount of phosphoric acid (13.91 mg g^−1^ DW), which dramatically (fourfold) decreased during the next five days and later on remained at a low level (3.65–1.44 mg g^−1^ DW). The content of phosphorus (P) in phosphoric acid at the 34th DAS was lower than the content of inorganic phosphorus (Pi) in a shoot of spelt wheat at the comparable developmental stage, as documented earlier [60]. Changes in the Pi content in shoots of wheat depend on P uptake from the soil, and plants’ P demand is higher at early rather than later stages of vegetation [61]. Results of our previous study, focusing on metabolomic changes in pea shoots (stems, stipules, tendrils, and shoot tips), revealed a decrease in phosphoric acid in shoot tips and tendrils, while there was an increase in stems and stipules during vegetative plant development (between the 35th and 53rd DAS) [33]. However, data on changes in phosphoric acid in wheat during plant vegetation are not known.

### 2.3. Changes in Polar Metabolite Profiles under Short-Term Soil Drought and Followed by Rewatering

The applied experimental conditions enabled us to characterize the metabolic adjustment of wheat to the short-term periodic water deficit manifested by the plants wilting. Results from the PCA score plots indicate a clear separation of control from droughted plants (Figure 4A). There was also a clear separation of samples from the first and second drought stresses, as well as a much lower separation of the control samples (by PC2, Figure 4A). The loading plots revealed that the discrimination of samples by PC1 was mainly due to sucrose, fructose, glucose, and malic acid, whereas phosphoric acid and proline contributed to the separation of samples by PC2. 

The ability of plants to survive drought stress depends on maintaining turgor (observed in our study during recovery, Figure 2) and is possible, e.g., by closing the stomata, limiting transpiration [11], and accumulating various osmolytes in the cytoplasm and vacuoles—sugars, amino acids, but also inorganic ions and organic acids that lower the water potential of cells [12]. Indeed, the reaction of the shoots of wheat to the first and second soil droughts was an increase in the concentration of TIPMs (Table 1) to levels significantly (*p* < 0.05) higher after both the first and second droughts than before the drought or after rewatering. Moreover, the concentration of TIPMs was significantly (*p* < 0.05) higher after the second soil drought than after the first one (Table 1). It was a result of the accumulation of soluble carbohydrates, increasing from 12.05 to 59.18 and 64.12 mg g^−1^ DW (Table 1), mainly sucrose, glucose, fructose, and to a lesser extent 1-kestose (Figure 5A–D). 

Those sugars, as osmolytes, participate in maintaining cell water homeostasis [62]. However, in the same type of experiment but performed on pea shoots [34], such changes were not observed. Moreover, different reactions to drought in terms of sucrose accumulation were also described in other species; e.g., the level of sucrose in the leaves of lentils decreased [63], while it increased in the leaves of maize [64], soybean [31], and wheat [39]. The accumulation of sucrose may also be influenced by the genetically determined resistance of plants to stress—in drought-resistant varieties, the level of sucrose increases, as was documented in soybean [65], chickpeas [66], barley [30], several species of *Lotus* [67], and *Triticeae* [68]. In our study, an accumulation of sucrose and 1-kestose (Figure 5A,D) in shoots of wheat under drought was accompanied by an increase in glucose and fructose (Figure 5B,C)—products of sucrose hydrolysis by invertases [69]. So far, a significant effect of the accumulation of sucrose and fructans on the increased resistance of plants to soil drought has been demonstrated in chicory [70] and ryegrass [71].

Among proteinogenic amino acids, there was an increase in the content of proline (Figure 5E), to a level fourfold higher during the second drought (8 mg g^−1^ DW) than during the first one. The level of proline dramatically decreased after each plant’s rewatering. The results of our research are consistent with those previously obtained in metabolomic studies of leaves of wheat [11] and leaves, stems, stipules, tendrils, and shoot tips of pea [32,33,34] and confirm the well-documented accumulation of proline in the response of plant vegetative tissues to abiotic stresses [72,73]. Proline increases the resistance of plants to drought, which was demonstrated, for example, in wheat [11,28], some model species, like *Arabidopsis thaliana* [25] and *Medicago truncatula* [74], and in transgenic wheat [75] and soybean [76]. The protective properties of proline result from its participation in osmotic regulation, stabilization of cellular structures (i.e., membranes and proteins), scavenging free oxygen radicals [77], and maintaining the intracellular redox potential by regulating the correct ratio of NADP^+^/NADPH in the cytosol, as well as the ability of proline to bind ^1^O_2_ [72,73].

Additionally, a significant increase in the content of leucine, isoleucine, and valine (branched-chain amino acids—BCAAs), but to a level much lower than that of proline, was found after the second soil drought (Table 1). The increase in those amino acids in wheat leaves was revealed earlier in both drought-tolerant and susceptible wheat cultivars [11,28]. 

Among the remaining polar metabolites, the content of malic acid also increased in the tissues under drought and decreased after rewatering. However, after rewatering, its concentration was still at a much higher level than in the control (Figure 5F). The accumulation of malate in response to drought was found earlier in the leaves of six wheat genotypes [39], as well as in shoots [33] and leaves of pea [32]. Malate participates in the osmotic regulation of guard cells, maintains the pH of the cytoplasm, and provides NADH for nitrate reduction [78].

The comparison of metabolite profiles of samples after drought stress and followed by recovery revealed the separation of samples according to the PC1 sharing 84.5% of variance (Figure 6A). 

Samples of drought-stressed plants (D1 and D2) were grouped on the right side of the PC1, whereas samples after recovery (R1 and R2) appeared on the left side (Figure 6A). The alteration in metabolic profiles was also observed following exposure to each drought stress treatment due to PC2. Moreover, the PC2 separated each recovery (Figure 6A). Such a distribution of samples resulted from the changes in the content of sucrose, fructose, glucose, proline, and phosphoric acid in particular (Figure 6B). Moreover, HCA separated samples into several clusters (Figure 6C). Apart from the clear separation of droughted and rewatered samples revealed by PCA, there were also control samples in one cluster shown by HCA. Additionally, rewatered samples were more closely related to the initial samples (in one cluster with samples from the 14th DAS) than to appropriate controls (in the 24th and 34th DAS).

The ability of plants to restore metabolism after the stress has subsided and to modify their behavior in response to a subsequent stress seems crucial due to the increasing intensity of environmental disturbances, including the predicted further global warming, which may be associated with an increased risk of periodic drought [79]. Our study revealed that the concentration of polar metabolites in samples after each recovery to optimal hydration, in relation to control samples (at the 24th and 34th DAS), was significantly higher. This was mainly due to the persistently elevated levels of fructose, glucose, sucrose, malic acid, and phosphoric acid after the first rewatering cycle and sucrose, malic acid, and phosphoric acid after the second cycle (Table 1). 

## 3. Materials and Methods

### 3.1. Material

Kernels of wheat (*Triticum aestivum* L. cv. Forkida, winter cultivar) were purchased from a domestic seed company—DANKO Plant Breeding. The experiment was performed as previously published on peas [34]. Briefly, kernels (after surface decontamination) were sown in plastic seedling trays (32 × 32 × 5 cm), three kernels per each of 25 cells, filled with 70 cm^3^ of garden soil (substral osmocote). The soil moisture was kept at 70–75% field water capacity (FWC), measured using a soil moisture meter (Theta Probe ML3, Delta-T, UK). Plants were cultivated in a greenhouse laboratory in April and May 2017. Soil drought was caused by the cessation of watering at the early stage of vegetative growth (from the 14th DAS)—at the stage when three leaves formed. After five days, the FWC decreased to 20–25%, and then watering resumed for 5 days. The watering cessation followed by rewatering (for 5 days each period) was repeated. Control plants grew at optimal FWC (70–75%) until the 34th DAS. 

### 3.2. Methods

Shoots from control wheat plants and those of two cycles of soil drought/rewatering were collected at the 14th, 19th, 24th, 29th, and 34th DAS, always between 9 and 10 a.m. (in 4 replicates), weighed, and frozen in liquid nitrogen. The time elapsed between the shoot harvesting and tissue freezing was no longer than 10 min. Samples were stored in an ultra-refrigerator (for 7 days at −76 °C) and freeze-dried for 48 h (shelf freeze-dryer, Alpha 1–2 LD, Martin Christ, Osterode am Harz, Germany). The WC was expressed in g of water per g of dry weight and calculated as WC (g g^−1^ DW) = (FW – DW)/DW [80].

#### 3.2.1. Analysis of Polar Metabolites

The extraction of polar metabolites was carried out according to the method described earlier [33,81]. The polar metabolites were extracted from 40 mg of dry (freeze-dried) tissues of wheat shoots (from 4 biological replicates) with a mixture of methanol:water. Homogenates were centrifuged, and aliquots of the clear supernatant were mixed with cold chloroform to remove non-polar compounds. The polar fraction was concentrated to dryness in a speed vacuum rotary evaporator (JW Electronic, Warsaw, Poland). The metabolites were derivatized with *O*-methoxamine hydrochloride and a mixture of MSTFA (*N*-methyl-*N*-trimethylsilyl-trifluoroacetamide) with pyridine. The mixtures of TMS (trimethylsilyl) derivatives were separated on a ZEBRON ZB-5MSi Guardian capillary column (Phenomenex, Torrance, CA, USA). Metabolic profiling of tissues was performed with the gas chromatography (GC) technique coupled with mass spectrometry (GC-MS), using a QP-GC-2010 apparatus (Shimadzu, Tokyo, Japan). Polar metabolites were identified and characterized by comparison of their retention time (RT), retention indices (RI, determined according to the saturated hydrocarbons), and mass spectra of original standards derived from Sigma-Aldrich (Sigma-Aldrich, Merck, Rahway, NJ, USA), and from the NIST library (National Institute of Standards and Technology). 

#### 3.2.2. Statistics

The results were subjected to a one-way ANOVA with a post-hoc test (Tukey) or Student’s *t*-test using Statistica software (version 12.0). Graphs were prepared using GraphPad Prism (version 3.0). Principal Component Analysis (PCA) and hierarchical cluster analysis (HCA) were performed in the COVAIN program [82], using the MATLAB software (version 2013a, Math Works), in order to compare the metabolic profiles of wheat during vegetative growth as well as under soil drought and two cycles of soil drought/rewatering. In addition, we also performed a pathway analysis of differentially accumulated polar metabolites after wheat plants exposure to drought in comparison to controls. It was performed using the MetaboAnalyst 5.0 platform (https://www.metaboanalyst.ca/MetaboAnalyst/ModuleView.xhtml, accessed on 1 March 2023). Although the *p*-value from the enrichment analysis was less than 0.05 for several pathways, there was a very low match status of metabolites for each of the pathways (Appendix A, Appendix A).

## 4. Conclusions

Results of the present study revealed dynamic changes in primary polar metabolites during the plants exposure to repeated short-term soil drought and rewatering. Soluble carbohydrates (sucrose, glucose, fructose, and 1-kestose), proline, and malate participate in this process due to their increasing concentration in response to drought and decreasing after rewatering. It was confirmed by repeated drought/rewatering cycles. However, the possible changes in the pattern of sugar (and other photoassimilates) translocation between shoots and roots under drought need further studies. The research also shows that GC-MS is an appropriate approach for fast screening of metabolic disturbances in primary metabolism in plants exposed to water deficit stress. However, for a deeper explanation of wheat adaptation to drought, further research focusing on prolonged periodic droughts as well as examining root tissues is needed.

## Figures and Tables

**Figure 1 ijms-24-08429-f001:**
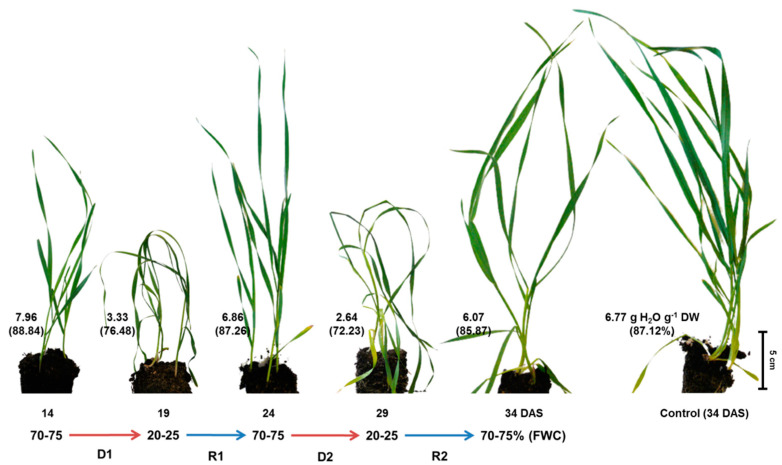
Plants of wheat (*Triticum aestivum* L., cv. Forkida) during repeated short-term soil drought during 14–19 DAS and 24–29 DAS and subsequent rewatering (19–24 and 29–34 DAS). Control plants at 34 DAS are shown on the right. The values in the bottom left side of each picture indicate the water content expressed in g H_2_O on g of dry weight and as % of fresh weight (in parentheses) in the shoot. Abbreviations: D1 and D2—first and second soil drought; R1 and R2—rewatering; FWC—field water capacity.

**Figure 2 ijms-24-08429-f002:**
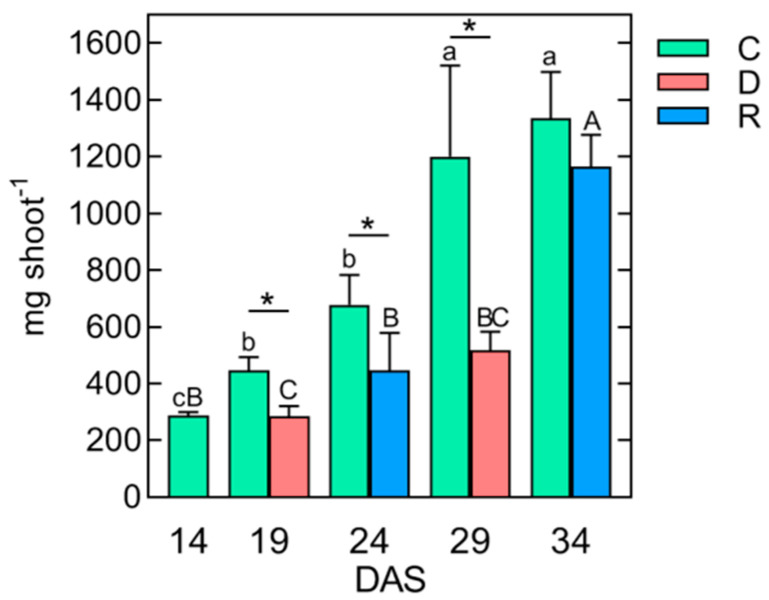
Changes in the fresh weight of the wheat (*Triticum aestivum* L., cv. Forkida) shoot during plant growth at control conditions (C) or exposed to short-term soil droughts (D, 14–19 and 24–29 DAS) and rewatering (R, 19–24, and 29–34, respectively). Values are means (*n* = 4) + SD. The same letters above the bars indicate no statistically significant differences (*p* < 0.05) based on ANOVA and the Tukey post-hoc test for control plants (a–c, 14–34 DAS) or plants before and after soil droughts and followed by recoveries (A–C). The significant differences in FW between shoots after 5 days of drought or followed by rewatering plants and control plants (from the same day after sowing, DAS) were marked with an asterisk, based on the Student’s *t*-test (*, *p* < 0.05).

**Figure 3 ijms-24-08429-f003:**
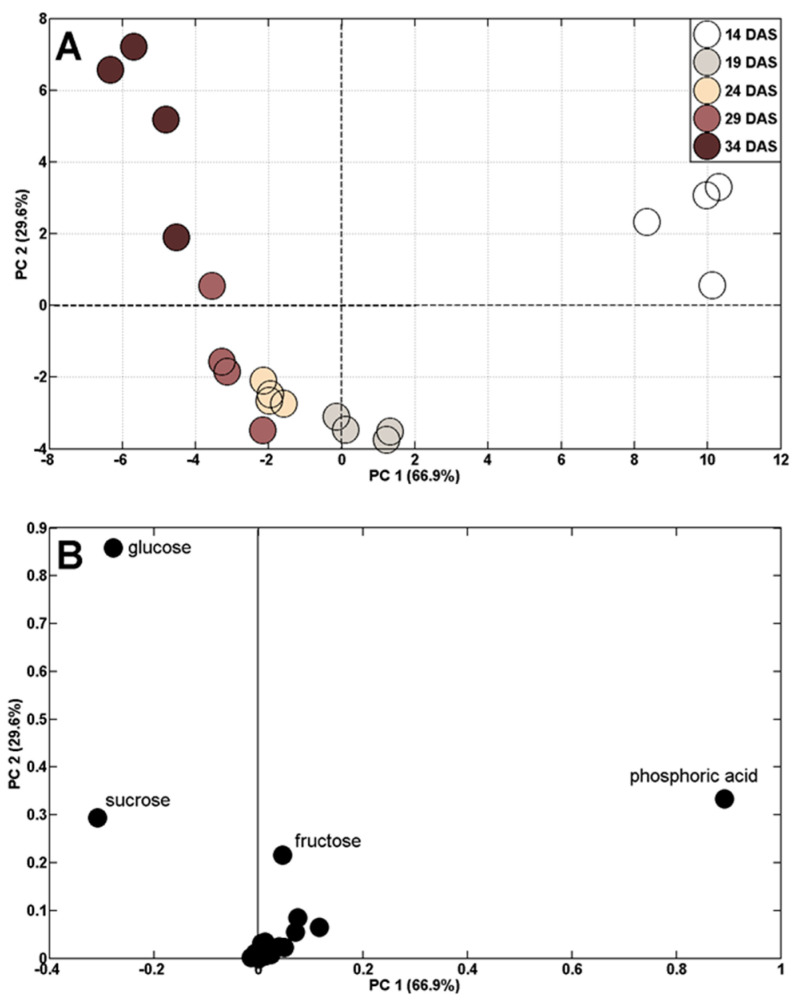
PCA of metabolite profiles of wheat shoots during plant vegetation (14–34 DAS) at control conditions (**A**) and PCA loading plots of polar metabolites (**B**).

**Figure 4 ijms-24-08429-f004:**
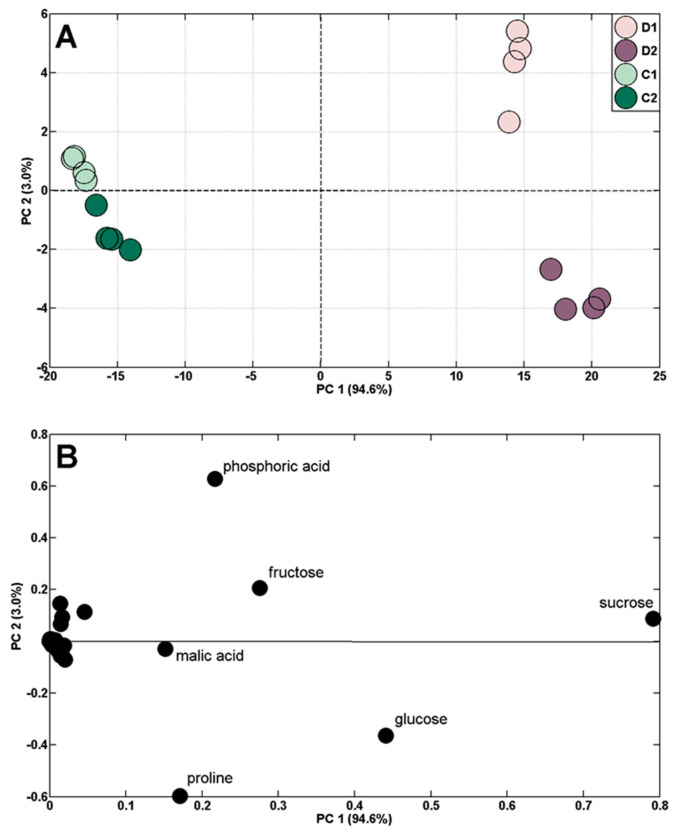
PCA of metabolite profiles of shoots of control (C1 and C2) and droughted (D1 and D2) plants (samples collected at 19 and 29 DAS) (**A**). PCA loading plots of polar metabolites are shown in (**B**).

**Figure 5 ijms-24-08429-f005:**
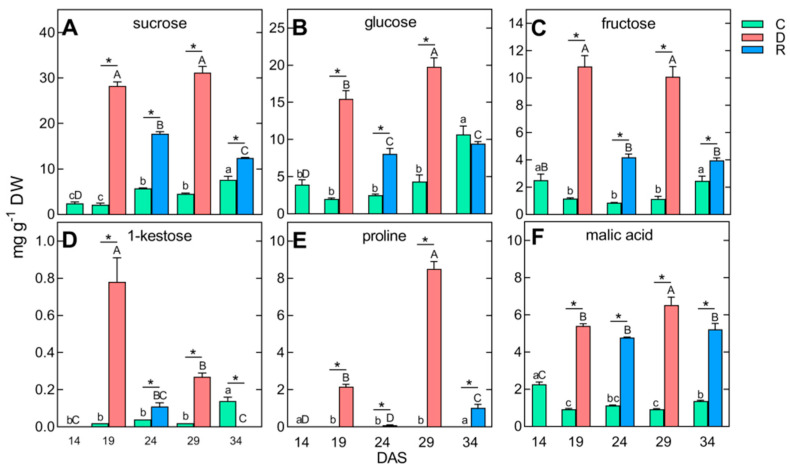
Changes in the concentration of selected polar metabolites, whose content increased under drought and decreased after rewatering (in both cycles) in wheat shoots: sucrose (**A**), glucose (**B**), fructose (**C**), 1-kestose (**D**), proline (**E**), and malic acid (**F**). Values are means (*n* = 4) ± SD. Abbreviation as in Figure 2.

**Figure 6 ijms-24-08429-f006:**
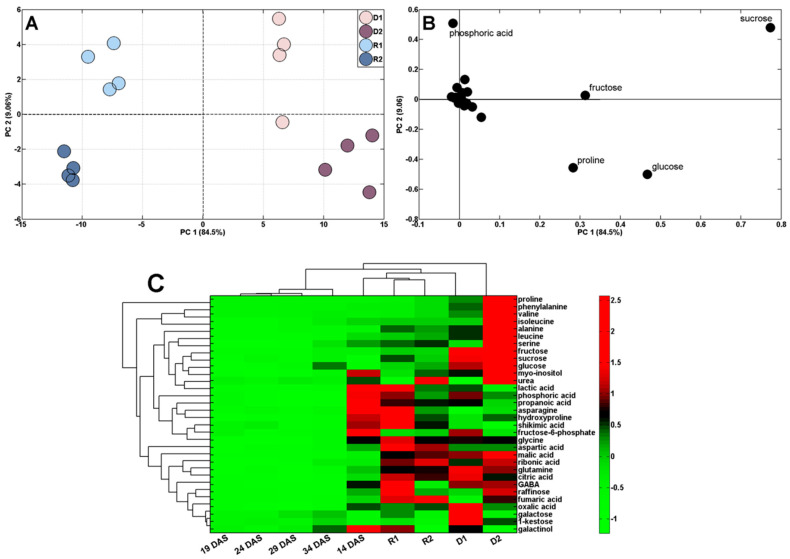
PCA of metabolite profiles of shoots of droughted (D1 and D2; samples collected at the 19th and 29th DAS) and rewatered (R1 and R2; samples collected at the 24th and 34th DAS) plants (**A**). PCA loading plots of polar metabolites are shown in (**B**). Hierarchical cluster analysis (HCA, **C**).

**Table 1 ijms-24-08429-t001:** The concentration (in mg g^−1^ DW) of polar metabolites in wheat shoots (*Triticum aestivum* L., cv. Forkida) during 20 days of plant vegetation (from the 14th to the 34th DAS) at optimal soil moisture (control), after 5 days of watering withdrawal from the 14th to the 19th and from the 24th to the 29th DAS (drought, D1 and D2, respectively) and followed by rewatering for 5 days, between the 19th and 24th and between the 29th and 34th DAS (R1 and R2, respectively). Statistical abbreviation as shown in Figure 2.

	Control					Drought (D)/Rewatering (R)			
DAS	14	19	24	29	34	D1 14–19	R1 19–24	D2 24–29	R2 29–34
Metabolites	mg g^−1^ DW								
TIPMs, including:	33.74 ^aE^	11.99 ^c^	14.47 ^c^	14.19 ^c^	26.96 ^b^	84.18 ^B^*	57.56 ^C^*	95.65 ^A^*	48.52 ^D^*
TSCs, including:	12.05 ^bE^	6.12 ^c^	9.75 ^bc^	10.84 ^b^	22.17 ^a^	59.18 ^B^*	33.09 ^C^*	64.12 ^A^*	28.20 ^D^*
fructose	2.50 ^aB^	1.16 ^b^	0.85 ^b^	1.14 ^b^	2.45 ^a^	10.85 ^A^*	4.18 ^B^*	10.09 ^A^*	3.97 ^B^*
galactose	0.00 ^aB^	0.04 ^a^	0.05 ^a^	0.18 ^a^	0.27 ^a^	1.21 ^A^*	0.37 ^B^	0.10 ^B^	0.07 ^B^
glucose	3.92 ^bD^	2.00 ^b^	2.53 ^b^	4.35 ^b^	10.70 ^a^	15.45 ^B^*	8.06 ^C^*	19.76 ^A^*	9.45 ^C^
sucrose	2.43 ^cD^	2.17 ^c^	5.73 ^b^	4.56 ^b^	7.63 ^a^	28.24 ^A^*	17.70 ^B^*	31.22 ^A^*	12.40 ^C^*
galactinol	0.10 ^aA^	0.03 ^c^	0.02 ^c^	0.03 ^c^	0.06 ^b^	0.07 ^B^*	0.08 ^AB^*	0.00 ^C^*	0.00 ^C^*
raffinose	0.53 ^aBC^	0.19 ^b^	0.23 ^b^	0.17 ^b^	0.17 ^b^	0.45 ^C^*	1.11 ^A^*	0.89 ^AB^*	0.52 ^BC^*
1-kestose	0.00 ^bC^	0.02 ^b^	0.04 ^b^	0.02 ^b^	0.14 ^a^	0.78 ^A^*	0.11 ^BC^*	0.27 ^B^*	0.00 ^C^*
*myo*-inositol	0.40 ^aA^	0.09 ^bc^	0.06 ^c^	0.09 ^bc^	0.15 ^b^	0.32 ^B^*	0.20 ^C^*	0.46 ^A^*	0.29 ^B^*
ribonic acid	0.40 ^aC^	0.11 ^b^	0.17 ^b^	0.20 ^b^	0.39 ^a^	0.66 ^B^*	0.80 ^A^*	0.87 ^A^*	0.91 ^A^*
fructose-6-phosphate	1.77 ^aA^	0.31 ^b^	0.08 ^b^	0.09 ^b^	0.21 ^b^	1.14 ^B^*	0.49 ^C^*	0.46 ^C^*	0.58 ^C^*
TPAAs, including:	2.18 ^aC^	0.42 ^c^	0.56 ^c^	0.60 ^c^	1.05 ^b^	4.27 ^B^*	2.78 ^C^*	13.40 ^A^*	3.33 ^BC^*
alanine	0.05 ^aB^	0.02 ^a^	0.02 ^a^	0.04 ^a^	0.03 ^a^	0.13 ^B^*	0.12 ^B^*	0.26 ^A^*	0.11 ^B^*
asparagine	0.68 ^aA^	0.10 ^b^	0.13 ^b^	0.10 ^b^	0.14 ^b^	0.11 ^C^	0.67 ^A^*	0.25 ^BC^*	0.33 ^B^*
aspartic acid	0.29 ^aC^	0.05 ^c^	0.10 ^bc^	0.09 ^bc^	0.13 ^b^	0.31 ^C^*	0.58 ^A^*	0.30 ^C^*	0.44 ^B^*
glutamine	0.05 ^aA^	0.01 ^c^	0.02 ^c^	0.01 ^c^	0.03 ^b^	0.10 ^A^*	0.08 ^A^*	0.09 ^A^*	0.08 ^A^*
glycine	0.24 ^aB^	0.06 ^c^	0.07 ^c^	0.06 ^c^	0.11 ^b^	0.24 ^B^*	0.30 ^A^*	0.24 ^B^*	0.24 ^B^*
isoleucine	0.20 ^aB^	0.04 ^c^	0.05 ^c^	0.08 ^c^	0.15 ^b^	0.21 ^B^*	0.21 ^B^*	0.78 ^A^*	0.22 ^B^*
leucine	0.12 ^aB^	0.02 ^b^	0.03 ^b^	0.03 ^b^	0.04 ^b^	0.22 ^B^*	0.14 ^B^*	0.50 ^A^*	0.17 ^B^*
phenylalanine	0.03 ^aC^	0.01 ^b^	0.01 ^b^	0.01 ^b^	0.01 ^b^	0.16 ^B^*	0.03 ^C^*	0.53 ^A^*	0.07 ^C^*
proline	0.04 ^aD^	0.02 ^b^	0.01 ^b^	0.01 ^b^	0.01 ^b^	2.17 ^B^*	0.09 ^D^*	8.50 ^A^*	1.02 ^C^*
serine	0.43 ^aB^	0.09 ^b^	0.10 ^b^	0.14 ^b^	0.33 ^a^	0.34 ^B^*	0.48 ^B^*	0.93 ^A^*	0.52 ^B^*
valine	0.05 ^abC^	0.01 ^c^	0.01 ^c^	0.04 ^bc^	0.07 ^a^	0.27 ^B^*	0.08 ^C^*	1.01 ^A^*	0.15 ^C^*
NTPAAs, including:	1.61 ^aAB^	0.32 ^b^	0.34 ^b^	0.35 ^b^	0.47 ^b^	0.98 ^C^*	2.02 ^A^*	1.27 ^BC^*	1.14 ^BC^*
hydroxyproline	1.39 ^aAB^	0.27 ^b^	0.31 ^b^	0.32 ^b^	0.43 ^b^	0.70 ^C^*	1.69 ^A^*	0.99 ^BC^*	1.04 ^BC^*
GABA	0.22 ^aB^	0.05 ^b^	0.04 ^b^	0.03 ^b^	0.04 ^b^	0.27 ^AB^*	0.34 ^A^*	0.28 ^AB^*	0.10 ^C^*
TOAs, including:	3.96 ^aB^	1.47 ^bc^	1.60 ^bc^	1.26 ^c^	1.82 ^b^	8.10 ^A^*	7.66 ^A^*	8.43 ^A^*	7.62 ^A^*
citric acid	0.96 ^aB^	0.33 ^b^	0.28 ^b^	0.23 ^b^	0.29 ^b^	2.18 ^A^*	2.06 ^A^*	1.55 ^AB^*	1.77 ^AB^*
fumaric acid	0.07 ^aC^	0.04 ^b^	0.04 ^b^	0.03 ^b^	0.04 ^b^	0.05 ^C^*	0.14 ^AB^*	0.11 ^B^*	0.15 ^A^*
malic acid	2.26 ^aC^	0.93 ^c^	1.13 ^bc^	0.92 ^c^	1.36 ^b^	5.39 ^B^*	4.77 ^B^*	6.53 ^A^*	5.22 ^B^*
oxalic acid	0.06 ^aB^	0.02 ^b^	0.01 ^c^	0.01 ^c^	0.02 ^b^	0.13 ^A^*	0.06 ^B^*	0.06 ^B^*	0.07 ^B^*
lactic acid	0.13 ^aAB^	0.03 ^b^	0.02 ^b^	0.02 ^b^	0.03 ^b^	0.09 ^ABC^*	0.14 ^A^*	0.05 ^C^*	0.08 ^BC^*
propanoic acid	0.23 ^aA^	0.06 ^b^	0.04 ^bc^	0.02 ^c^	0.04 ^bc^	0.15 ^B^*	0.16 ^B^*	0.10 ^C^*	0.14 ^BC^*
shikimic acid	0.24 ^aB^	0.07 ^b^	0.09 ^b^	0.03 ^c^	0.05 ^c^	0.11 ^C^	0.34 ^A^*	0.03 ^D^	0.19 ^B^*
TRCs, including:	13.94 ^aA^	3.65 ^b^	2.21 ^c^	1.14 ^d^	1.45 ^cd^	11.65 ^A^*	12.00 ^A^*	8.43 ^B^*	8.23 ^B^*
phosphoric acid	13.93 ^aA^	3.65 ^b^	2.21 ^c^	1.13 ^d^	1.44 ^cd^	11.65 ^A^*	12.00 ^A^*	8.40 ^B^*	8.21 ^B^*
urea	0.01 ^aB^	0.00 ^a^	0.00 ^a^	0.01 ^a^	0.00 ^a^	0.00 ^C^	0.00 ^C^*	0.02 ^A^	0.02 ^A^*

Abbreviations: DAS—day after sowing; TIPMs—total identified polar metabolites; TPAAS—total proteinogenic amino acids; TNAAs—total non-proteinogenic amino acids; TOAs—total organic acids; TRCs—total remaining compounds.

## Data Availability

The data presented in this study are available in this article.

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
