# Peer review of "Polar Metabolites Profiling of Wheat Shoots (Triticum aestivum L.) under Repeated Short-Term Soil Drought and Rewatering"

_ijms, 2023, doi:10.3390/ijms24098429_

Round 1

Reviewer 1 Report (Previous Reviewer 1)

Authors have incorporated the minor suggestions and now the revised version can be accepted for publication in IJMS. 

Minor editing of English language required

Author Response

Thank you very much for your positive review. I think that minor errors in the English language will be removed after linguistic correction, which is carried out before publication.

Reviewer 2 Report (Previous Reviewer 3)

The abstract section needs improvement. It should provide a concise summary of the research article, including the methodology used for the evaluation. For instance, it would be helpful to mention the number of metabolites that were identified using GC-MS. Ambiguity in expression, such as the use of words like "seem," should be avoided in the abstract. Additionally, the sentence "The possible changes in the pattern of sugars (and other photoassimilates) translocation between shoots and roots under drought need further studies" should be moved to the conclusion section. Furthermore, it would be beneficial to include a discussion on the expectations or potential future directions resulting from this research.

Author Response

Thank you very much for your positive review.

The abstract has been changed as follows:

“The response of wheat (Triticum aestivum L.) plants to the soil drought at the metabolome level remains still not fully explained. In addition, research focuses mainly on single periods of drought, and there is still a lack of data on the response of plants to short-term cyclical periods of drought. The key to this research was to find whether wheat shoots are able to resume metabolism after the stress subsides and if the reaction to subsequent stress is the same. Gas chromatography coupled with mass spectrometry (GC-MS) is one of the valuable and fast methods to discover changes in primary metabolism of plants. The targeted GC-MS analyses of whole shoots of wheat plants exposed (at the juvenile stage of development) to short-term (five days) mild soil drought/rewatering cycles (until the start of shoots wilting) enabled us to identify 32 polar metabolites. The obtained results revealed an accumulation of sugars (sucrose, fructose, glucose, 1-kestose) proline and malic acid. During five days of recovery, shoots regained full turgor and continued to grow, and the levels of accumulated metabolites decreased. Similar changes in metabolic profiles were found during the second drought/rewatering cycle. However, the concentrations of glucose, proline and malic acid were higher after the second drought than after the first one. Additionally, the concentration of total polar metabolites after each plant rewatering was elevated, compared to control samples. Although our results confirm the participation of proline in wheat response to drought, they also highlight the responsiveness of soluble carbohydrates metabolism to stress/recovery.”

The conclusions have been changed as follows:

"Results of the present study revealed dynamic changes in primary polar metabolites during the plants exposure to repeated short-term soil drought and rewatering. Soluble carbohydrates (sucrose, glucose, fructose and 1-kestose), proline and malate participate in this process due to their increasing concentration in response to drought and decrease after rewatering. It was confirmed by repeated drought/rewatering cycle. However, the possible changes in the pattern of sugars (and other photoassimilates) translocation between shoots and roots under drought need further studies. The research shows also that GC-MS is an appropriate approach for fast screening of metabolic disturbances in primary metabolism in plants exposed to water deficit stress. However, for a deeper explanation of wheat adaptation to drought, further research focusing on prolonged periodic droughts, as well as to examine root tissues are needed."

Reviewer 3 Report (New Reviewer)

The manuscript "Polar metabolites profiling of wheat shoots (Triticum aestivum L.) under repeated short-term soil drought and rewatering" used GC-MS to measure the polar metabolites of wheat shoots. The experiments are well-designed and performed, and the conclusions are appropriate to the results presented. The current version looks good to me, and I don’t have any comments. 

Author Response

Thank you very much for your positive review.

This manuscript is a resubmission of an earlier submission. The following is a list of the peer review reports and author responses from that submission.

Round 1

Reviewer 1 Report

This study used the GC–MS method for polar metabolites profiling wheat shoots (Triticum aestivum L.) during exposure to repeated short-term soil drought and rewatering. The key to this research was to find whether wheat shoots can resume metabolism after the stress subsides and if the reaction to subsequent stress is the same. Drought is becoming one of the major abiotic stresses, and it greatly impacts wheat production worldwide. Hence, finding drought-related metabolites is an interesting approach for wheat metabolic engineering and advancement. However, the current draft demands some improvement.

The objectives and hypothesis are missing in the abstract.

Please use some diverse keywords that have not been used in the title.

Ref 6-8 need to be replaced with the most recent articles such as doi: 10.1002/tpg2.20279; and 10.1002/ggn2.202100017.

There are some undefined abbreviations in m&m and elsewhere. Please carefully check and define them on the first mention in the main text.

In 3.2.1., The polar metabolites were extracted from 40 mg of dry tissues. Add the tissue name within parentheses.

Why authors did not perform the functional annotation analysis such as KEGG.

Though the results and discussion are well-presented. Still, some parts need to be further improved, such as kindly telling us what the dataset reflects and not simply explaining the data. Also, explain the role of metabolites in drought tolerance in discussion. The discussion parts need to be further improved.

Reviewer 2 Report

This paper attempts to profile polar metabolites in wheat stems under drought and rewatering conditions. In itself, the paper performs a correct methodology to achieve the objectives of the paper. However, the paper suffers from being a properly scientific paper; apart from a determination of a large number of metabolites, there is no other result to explain why these substances are distributed. The authors simply speculate on the role that these metabolites might play on the basis of other works. The work is therefore highly descriptive.

Regarding the introduction, it is very short in length, and presents a high degree of plagiarism.

The material and methods are correct and well described.

However, as mentioned above, the discussion is unscientific, as there is no data to support the role that the analysed metabolites might play in drought stress. The authors simply support their speculation on the role of each of them in the literature.

One of the aims of the paper is to identify possible markers for drought stress, but the data are purely speculative. Undoubtedly, Proline levels are probably related to the drought processes applied, but there are other metabolites that cannot be known.

First of all, the authors should have measured the water potential, or even better, the solute potential, so that one could talk about the role of some metabolites as compatible osmolytes. Without these measurements it cannot be concluded that there is drought stress. The only measure you have is the water content, the value of which is not useful to compare between treatments; for that you should have calculated the relative water content. Undoubtedly, the decrease in water content must be accompanied by a loss of cell turgor, but as no measurements have been made to characterise this, we cannot know the level of turgor loss, and to which degree it affects the functioning of the plant.

Another important problem is that it has been measured in the stem, so some of these metabolites do not necessarily play an osmotic role, and are simply being transported via the phloem; in fact it is striking to note the large amount of sucrose and that the authors do not mention that it is the main substance that is transported via the phloem. The same authors comment that the role of sucrose in stress is not clear. These metabolites should have been measured in the leaf.

What is the role of phosphoric acid? It is only mentioned that it increases, but nothing else is mentioned.

The clearest summary of all this can be seen in the conclusions of the paper:

"Soluble carbohydrates (sucrose, glucose, fructose and 1-kestose) and proline seem to play a key role in this process due to their increasing concentration in response to drought and decrease after rewatering cycles".

These metabolites seem to play a role, but nothing more can be clarified because of the descriptive style of the paper.

There is no data beyond the relationship between the decrease of water content and the increase of some metabolites. 

Nor can it be concluded that all of them are markers for drought stress, as the level of drought stress has not been characterised, nor can it be concluded that all of them are increased on the basis of drought stress.

Author Response

Dear reviewer,

We sincerely thank you for your comments on our manuscript. We have provided answers to your comments below.

Q1:

Regarding the introduction, it is very short in length, and presents a high degree of plagiarism.

Answer:

The introduction has been edited and the following parts have been added:

“Drought adversely affects crop growth, development and yield via physiological and biochemical disturbances in whole plants. However, the negative effects are related to the drought intensity (rate of decrease in the water availability and water deficit duration), as well as plant species and developmental stage [7,8].”

“Water stress leads to weakening and inhibition of plant growth, disturbances in photosynthesis and carbon and nitrogen metabolism [9] as well as overproduction of reactive oxygen species [10]. The limited and decreasing availability of water in the soil, leading to soil drought, indicates changes in stomata aperture in leaves to limit transpiration and water loss from the cells [11]. Water retention in cells is achieved through the accumulation of various osmolytes in the cytoplasm and vacuole – sugars, amino acids, inorganic ions and organic acids, lowering the water potential [12]. However, prolonged water deficit and minimized gas exchange lead to a decrease in CO2 concentration in the leaf [13,14]. The decreased ratio of CO2 to O2 in chloroplasts results in the intensification of the photorespiration process [15,16]. It reduces the efficiency of CO2 assimilation and causes losses of nitrogen and energy. However, it can also be a partial CO2 recirculation pathway [17,18]. Limited water availability and reduced CO2 assimilation (in the Calvin-Benson cycle) lead to disturbances in the processes of electron transport, energy production and maintaining redox balance in chloroplasts. As a result, excessive generation of reactive oxygen species and the photosynthetic apparatus is damaged [19]. This causes disturbances in the course of various metabolic pathways and the export of photoassimilates from the leaves (as sources) to the receiving sink tissues (roots, stem, generative organs) [20,21].”

“So far, research on wheat's response to drought at the metabolome level has shown that limited water availability is associated with changes in osmotic regulation, energy production and oxidative damages [37,38]. It has been demonstrated that sugars and nitrogen metabolism are mostly affected by water deficiency in leaves of wheat [28,37-39]. Moreover, the drought-tolerant capacity of wheat seedlings was related to the accumulation of amino acids, organic acids, alkaloids and flavonoids [40]. The important role of some metabolites in response to drought has been also revealed by metabolomic analyses carried out on other plant species (i.e. coffee, ryegrass) subjected to cyclic drought/rehydration treatment [8,41,42]. In wheat, it has been shown that not only the water deficit itself but also its frequency affects plant growth and secondary metabolism in flag leaf [43] as well as that the greatest changes are related to the accumulation of amino acids [11] and soluble sugars in leaf tissues [44]. Moreover, the drought tolerance of wheat seems to be related to sucrose metabolism [45,46].

Among various methods used in metabolomic studies [30,31], the gas chromatography-mass spectrometry (GC–MS) method seems to be valuable in monitoring changes in metabolic profiles in early response to drought [11,27,33,34]. In the present study, GC-MS approach enabled us to compare changes in polar metabolites profiles of wheat shoots (Triticum aestivum L.) during the plant’s exposure to repeated short-term (mild) soil drought and after recovery from water deficit stress.”

Q2:

However, as mentioned above, the discussion is unscientific, as there is no data to support the role that the analysed metabolites might play in drought stress. The authors simply support their speculation on the role of each of them in the literature. The authors simply support their speculation on the role of each of them in the literature.

Q3:

One of the aims of the paper is to identify possible markers for drought stress, but the data are purely speculative. Undoubtedly, Proline levels are probably related to the drought processes applied, but there are other metabolites that cannot be known.

Answer:

All speculative suppositions has been removed.

The discussion has been edited and the following parts have been added:

“The content of phosphorus (P) in phosphoric acid at the 34th DAS was lower than the content of inorganic phosphorus (Pi) in a shoot of spelt wheat at the comparable developmental stage, documented earlier [60]. Changes in the Pi content in shoots of wheat depend on P uptake from the soil and plant’s P demand is higher at early than later stages of vegetation [61]. Results of our previous study, focusing on metabolomic changes in pea shoots (stems, stipules, tendrils and shoot tip) revealed a decrease in phosphoric acid in shoot tips and tendrils, while the increase in stems and stipules during vegetative plant development (between the 35th and 53rd DAS) [33]. However, data on changes in phosphoric acid in wheat during plants vegetation are not known.”

“Those sugars, as osmolytes, participate in maintaining cell water homeostasis [62]. However, in the same type of experiment but made for pea shoots [34], such changes were not observed. Moreover, different reactions to drought, in terms of sucrose accumulation, were also described in other species, e.g. the level of sucrose in leaves of lentil decreased [63], while it increased in leaves of maize [64], soybean [31] and wheat [39]. The accumulation of sucrose may also be influenced by the genetically determined resistance of plants to stress – in drought-resistant varieties, the level of sucrose increases, as was documented in soybean [65], chickpeas [66], barley [30], several species of Lotus [67] and Triticeae [68]. In our study, an accumulation of sucrose and 1-kestose (Figure 5A and D) in shoot of wheat under drought was accompanied by an increase in glucose and fructose (Figure 5B and C) – products of sucrose hydrolysis by invertases [69]. So far, a significant effect of the accumulation of sucrose and fructans on the increased resistance of plants to soil drought has been demonstrated in chicory [70] and ryegrass [71].”

“The accumulation of malate in response to drought was found earlier in leaves of six wheat genotypes [39] as well as in shoots [33] and leaves of pea [32]. Malate participates in osmotic regulation of guard cells, maintains pH of cytoplasm and provides NADH for nitrate reduction [78].”

“Samples of drought-stressed plants (D1 and D2) were grouped on the right side from the PC1, whereas samples after recovery (R1 and R2) appeared on the left side (Figure 6A). The alteration in metabolic profiles was also observed following exposure to each drought stress treatment due to PC2. Moreover, the PC2 separated each recovery (Figure 6A). Such a distribution of samples resulted from the changes in the content of sucrose, fructose, glucose, proline and phosphoric acid in particular (Figure 6B). Moreover, HCA separated samples into several clusters (Figure 6C). Besides clear separation of droughted and rewatered samples, revealed by PCA, there were also control samples in one cluster shown by HCA. Additionally, rewatered samples were more closely related to the initial samples (in one cluster with samples from the 14th DAS) than to appropriate controls (in the 24th and 34th DAS).”

Q4:

First of all, the authors should have measured the water potential, or even better, the solute potential, so that one could talk about the role of some metabolites as compatible osmolytes. Without these measurements it cannot be concluded that there is drought stress. The only measure you have is the water content, the value of which is not useful to compare between treatments; for that you should have calculated the relative water content. Undoubtedly, the decrease in water content must be accompanied by a loss of cell turgor, but as no measurements have been made to characterise this, we cannot know the level of turgor loss, and to which degree it affects the functioning of the plant.

Answer:

Unfortunately, during the biometric measurements, we did not measure the water content in the shoots in full turgor because it could be done separately for each leaf and stem. In our study we focused on the metabolic profile analysis in the whole shoot.

However, in reference to your comment, we have decided to express the water content in the tissues not only as  percentage of FW but also as a water content on a dry-weight basis which measures the mass ratio between water and the dry mass in tissues (expressed as grams of water per gram of dry weight) based on the recommendation of Sun 2002 in a publication entitled ‘Methods for the study of water relations under desiccation stress’.

Q5:

Another important problem is that it has been measured in the stem, so some of these metabolites do not necessarily play an osmotic role, and are simply being transported via the phloem; in fact it is striking to note the large amount of sucrose and that the authors do not mention that it is the main substance that is transported via the phloem. The same authors comment that the role of sucrose in stress is not clear. These metabolites should have been measured in the leaf.

Answer:

In our study, we measured the content of polar metabolites not in the stem but in the whole shoots and did not divide them into leaves and stems. Therefore, unfortunately, we are unable to specify their content in the leaves separately.

Q6:

What is the role of phosphoric acid? It is only mentioned that it increases, but nothing else is mentioned.

Answer:

The answer is given in Q2

Q7:

The clearest summary of all this can be seen in the conclusions of the paper:

"Soluble carbohydrates (sucrose, glucose, fructose and 1-kestose) and proline seem to play a key role in this process due to their increasing concentration in response to drought and decrease after rewatering cycles".

 These metabolites seem to play a role, but nothing more can be clarified because of the descriptive style of the paper.

There is no data beyond the relationship between the decrease of water content and the increase of some metabolites. 

Nor can it be concluded that all of them are markers for drought stress, as the level of drought stress has not been characterised, nor can it be concluded that all of them are increased on the basis of drought stress.

Answer:

The present/corrected form of conclusion is as follows:

“Results of the present study revealed dynamic changes in primary polar metabolites during the plants exposure to repeated short-term soil drought and rewatering. Soluble carbohydrates (sucrose, glucose, fructose and 1-kestose), proline and malate participate in this process due to their increasing concentration in response to drought and decrease after rewatering. It was confirmed by repeated drought/rewatering cycle. Thus, the GC-MS method seems to be an appropriate approach for fast screening of metabolic disturbances in primary metabolism in plants exposed to water deficit stress. However, for a deeper explanation of wheat adaptation to drought it seems necessary to conduct research during prolonged periodic droughts, as well as to examine root tissues.”

Reviewer 3 Report

The authors studied the morphological and physiological changes of wheat (Triticum aestivum L.)under drought and subsequences rewatering. Furthermore, the authors performed the metabolites profiling in wheat shoot under drought and subsequences rewatering. Authors found the accumulation of glucose, proline and malic acid, and the amount of these metabolites was higher under the second soil drought than after the first one. This research likely gives the knowledges for researchers who are working in research area involved in environmental stress response to plants and agricultural research but this report has shortcoming that need to be addressed before this can be published in a journal. In particular, if the metabolites selected in this study are to be used as metabolite markers during drought stress, I think that the correlation between the degree of drought stress to plants and the amount of metabolites should be investigated more strictly.

It was found that the changes in water content in plants were dramatically altered by the drought treatment. The authors observed the accumulation of metabolites during the drought process. Although the authors considered that these metabolites are probably used to avoid the next drought stress, is it possible to see their effects of plant on drought stress? For example, is there a difference in water loss in the soil or plants during drought stress treatment between the 1st drought and the 2nd drought? Also, is the recovery rate after the 1st drought treatment different from the recovery rate after the 2nd drought treatment? The authors suggested that these metabolites could be used as metabolic markers during drought stress, but the results were somewhat unreliable.

When is the sampling time? I think the amount of metabolites varies throughout the day. Please provide as much detailed information as possible when using it as a marker.

Author Response

Dear reviewer,

We sincerely thank you for your comments on our manuscript. We have provided answers to your comments below.

Q1:

In particular, if the metabolites selected in this study are to be used as metabolite markers during drought stress, I think that the correlation between the degree of drought stress to plants and the amount of metabolites should be investigated more strictly.

It was found that the changes in water content in plants were dramatically altered by the drought treatment. The authors observed the accumulation of metabolites during the drought process. Although the authors considered that these metabolites are probably used to avoid the next drought stress, is it possible to see their effects of plant on drought stress? For example, is there a difference in water loss in the soil or plants during drought stress treatment between the 1st drought and the 2nd drought? Also, is the recovery rate after the 1st drought treatment different from the recovery rate after the 2nd drought treatment? The authors suggested that these metabolites could be used as metabolic markers during drought stress, but the results were somewhat unreliable.

Answer:

Indeed, our earlier suggestions that the conducted research allows the identification of metabolic markers of wheat response to soil drought stress can be considered too unfounded. That is why we have made a change - abandoning the recognition of metabolites accumulated in response to stress as stress markers, and focusing attention on GC-MS approach that enabled us to compare changes in polar metabolites profiles of wheat shoots (Triticum aestivum L.) during the plant’s exposure to repeated short-term (mild) soil drought and after recovery. We find out that some soluble carbohydrates (sucrose, glucose, fructose and 1-kestose), proline and malate participate in the response/adaptation of wheat to repeated drought/rewatering cycles.

Q2:

When is the sampling time? I think the amount of metabolites varies throughout the day. Please provide as much detailed information as possible when using it as a marker.

Answer:

We supplemented this information by stating that the samples were always collected between 9 and 10 a.m.

Round 2

Reviewer 1 Report

I agree with the author's idea for not adding KEGG in the main MS. I suggest adding 1-2 lines in the manuscript for KEGG and adding these figure/table in the supplementary data. This would help the readers to understand why this part was not added in the document. 

Reviewer 3 Report

The answer to my question is incorrect. The abstract has been completely revised, and I am not sure about the novelty of this research result. It is understandable that the molecular mechanisms based on plant physiology caused by drought stress may be unclear, but what were the specific research points and why did the authors perform this experiment? Please write them in the abstract.